# Pediatric Oral Health Online Education for Rural and Migrant Head Start Programs in the United States

**DOI:** 10.3390/ijerph21050544

**Published:** 2024-04-25

**Authors:** Francisco Ramos-Gomez, Stephanie Parkinson, Victor Garcia de Jesus, Jose A. Rios, Janni J. Kinsler

**Affiliations:** 1Section of Pediatric Dentistry, Division of Growth & Development, Los Angeles (UCLA) School of Dentistry, University of California, Los Angeles, CA 90095, USA; sparkinson@dentistry.ucla.edu (S.P.); vgarciadejesus@dentistry.ucla.edu (V.G.d.J.); jarios@dentistry.ucla.edu (J.A.R.); jnhaiem@dentistry.ucla.edu (J.J.K.); 2Venice Family Clinic, Los Angeles, CA 90291, USA

**Keywords:** online oral health education, rural populations, migrant populations, Spanish-speaking populations, access to oral care, early childhood caries, pediatric dentistry, shortage of dental work force, community health workers

## Abstract

Children living in rural and migrant areas in the United States disproportionately suffer from poor oral health. Additionally, there continues to be a shortage of pediatric dentists practicing in rural/migrant areas. The purpose of this formative research study was to assess whether staff, teachers and families from rural/migrant Head Start/Early Head Start (HS/EHS) programs in California were receptive to oral health online education workshops conducted by pediatric dental residents who were assisted by bilingual (English and Spanish) community oral health workers (COHWs). Our findings suggest that partnering pediatric dental residents with bilingual COHWs to educate HS/EHS teachers, staff and parents on oral health care in rural/migrant areas could result in a rewarding experience for pediatric dentists that might lead them to practice in these communities upon graduation from their residency program. Furthermore, the positive feedback received from the teachers, staff and parents who participated in the workshops indicates they were receptive to receiving oral health information related to oral health literacy from the dental providers and COHWs. COHWs can help increase access to oral health care by serving as a bridge between families and providers by relaying information in a cultural, linguistic and sensitive manner.

## 1. Introduction

Children living in rural and migrant areas in the United States (US) disproportionately suffer from poor oral health [1]. For example, children living in rural areas are more likely to report unmet dental needs and less likely to have visited the dentist in the past year [1]. Studies have also shown that rural children are less likely to receive evidence-based preventive services, such as fluoride treatment and dental sealants, when compared to children in urban populations [2,3]. The COVID-19 pandemic highlighted the consequences that lack of access to care can have on rural communities across the US. During the pandemic, children living in rural areas were less likely to have had one or more preventive medical and dental visits in the past 12 months (59.6 percent) than urban children (66.7 percent) at all income levels [4]. Oral health is also one of the greatest unmet health needs of migrant and seasonal farmworkers [5]. Migrant/seasonal farmworkers experience 150 to 300% more decayed teeth than their non-migrant/seasonal farmworker peers, and approximately half of children who are farmworkers have at least one and an average of three teeth with cavities [5]. Migrant/seasonal workers, most of whom are Spanish speaking, face many barriers to receiving dental care, including lack of transportation and insurance, rural residency, the threat of wage or job loss, language barriers and limited clinic hours [5,6]. In addition to these barriers, many migrant/seasonal workers lack basic oral health knowledge [5]. 

Untreated dental caries can negatively impact growth, speech development and confidence, and lead to painful infections that can potentially spread systemically. Students with toothaches are four times more likely to have low grade point averages (GPAs), and students without access to needed dental care are almost three times as likely to miss school [7,8]. According to the Health Resources and Services Administration (HRSA), two-thirds of the nation’s dental health professional shortage areas (D-HPSAs) are in rural areas [9,10,11]. Rural health clinics (RHCs) and federally qualified health centers (FQHCs) are major healthcare access points for many rural and migrant/seasonal farmworker communities. However, of the 46 million Americans living in rural areas in 2017, 17 million lived in counties without RHCs, and 15 million lived in rural counties without FQHCs [12,13]. Furthermore, some of these clinics do not offer dental care, and among those that do, some do not provide dental care to young children. Fewer dental specialists, including pediatric dentists, practice in rural communities [11]. The shortage of dental workforce may become more acute with time, as 42 percent of dentists currently practicing in rural counties are over age 55 and may be nearing retirement [14].

In California, 54% of kindergarteners and 70% of third graders have experienced dental caries, and nearly one-third of children have untreated dental caries [15]. Almost half (43%) of California’s children live at or near the poverty level. Poverty rates range from approximately 20% in the Bay Area, Sacramento area and the northern region to nearly 30% in Los Angeles County and the Central Coast [16]. Furthermore, the poverty rate is 42% among young children with non-English-proficient parents in California and 40% among children with immigrant parents [16]. San Bernadino, Fresno, Riverside and Los Angeles counties cover a very large geographic area of California, including numerous pockets of rural and migrant areas and have one of the largest populations of Hispanics/Latinos in California (53.7%, 53.6%, 49.7 and 48%, respectively) [17]. Even though the majority of low-income children in Los Angeles County have dental insurance coverage (mostly through Medi-Cal’s dental program) [15], many children do not receive dental care due to issues with transportation, co-payments or deductibles, trust of providers and language barriers. Furthermore, only 58% of Los Angeles County residents have access to optimally fluoridated water, and those without it typically live in underserved neighborhoods, exacerbating oral health inequity [8,15,18,19,20,21].

As of 2017, California was ranked first as having the largest number of active dentists working; however, the distribution of where they practice is very lopsided [22]. They tend to cluster in areas such as San Francisco and Southern California, with very few practicing in rural/migrant and other underserved areas [22,23,24]. The University of California, Los Angeles (UCLA) School of Dentistry’s Section of Pediatric Dentistry works hard to recruit and train residents who express a passion for serving vulnerable populations. Between 2012 and 2023, 95 pediatric dental residents were trained to provide preventive, culturally competent and minimally invasive oral health care to underserved and special needs populations. A detailed description of the program can be found in previous publications [25,26,27,28]. The program, funded through a grant from the Health Resources and Services Administration (HRSA), which is currently in its third five-year cycle, recruits 7 to 10 residents every academic year. Over one-third of our pediatric dental residents (35%) come from a disadvantaged background and are considered underrepresented minorities, and 15% come from a rural background. Of the 53 alumni who responded to our alumni survey, 16 (34%) reported working in a medically underserved area. 

The use of community health workers (CHWs) or promotoras (lay-person health educators in the Hispanic/Latino community) can help increase access to oral health care by serving as a bridge between families and providers [29,30,31,32,33]. CHWs can help facilitate culturally and linguistically appropriate oral care that is healing, transformative and essential for addressing critical gaps in access to oral health care, especially in underserved and minority populations. They also play a key role in the development and implementation of community partnerships that result in positive health outcomes, as they possess an intimate understanding of the needs of their respective communities and can often provide the community members they serve with the most successful and cost-efficient routes to care. Bilingual (English and Spanish) pediatric dental residents from the UCLA School of Dentistry’s Center for Children’s Oral Health (UCCOH) conducted a series of oral health education training workshops with CHWs and Latina caregivers from a local Early Head Start (EHS) program in Los Angeles County. The study showed the oral health training workshops significantly improved knowledge, attitudes and self-reported practices of the CHWs and caregivers trained. Previous studies have also shown that oral health education programs conducted by CHWs can be an effective way to increase oral health knowledge in migrant populations [5,6]. However, to our knowledge, there is currently no published research that examines partnering pediatric dentists with community oral health workers (COHWs) to educate individuals who live in rural/migrant areas on oral health. Bringing dentists and COHWs or promotoras together is an innovative approach to building a collaborative relationship between dental providers and community-based organizations in rural/migrant areas as a way of promoting optimal oral health and whole-patient care.

The purpose of this formative qualitative research study was to assess whether staff, teachers and families from rural/migrant Head Start (HS) and EHS programs in California were receptive to oral health online education workshops conducted by pediatric dental residents who were assisted by bilingual (English and Spanish) COHWs as a dual strategy for (1) increasing the number of trained dentists who may choose to practice in underserved areas, and (2) improving access to care for underserved and hard-to-reach populations.

## 2. Materials and Methods 

### 2.1. Study Participants and Procedures

UCLA pediatric dental residents conducted 9 oral health online education workshops with the Riverside County Office of Education (RCOE) Rural/Migrant HS/EHS program and the Community Action Partnership of San Luis Obispo County (CAPSLO) between 23 August 2022 and 23 May 2023. The pediatric dental residents were assisted by bilingual (English and Spanish) COHWs who were available to help with issues related to health literacy. The workshops were conducted with a total of 451 teachers, staff and parents (276 teachers/staff and 175 parents). Four workshops were conducted with teachers and staff. These workshops were conducted in English and lasted approximately two hours. Five workshops were conducted with parents. These workshops were conducted in both English and Spanish and lasted between 45 min and one hour. Most workshops included 15 to 39 participants. However, one workshop for the parents included 60 participants and one workshop for the teachers and staff included 170 participants. All the workshops were conducted via Zoom due to COVID-19 restrictions.

### 2.2. Riverside County Office of Education (RCOE) Rural/Migrant Head Start/Early Head Start Program

The RCOE Rural/Migrant HS/EHS Program, in partnership with parents and community partners, facilitates access to comprehensive services for eligible migrant/seasonal farmworker children and their families. They provide a foundation for early childhood education, parental awareness and training on child health, growth and development. They aim to ensure an environment of responsive care-giving that leads to school readiness [34]. Currently, RCOE Rural/Migrant HS/EHS includes five early childcare centers (with 15 preschool classrooms and 10 toddler classrooms) and 54 family childcare home partnerships, and employs 40 certificated teachers, 15 instructional assistants and 22 certified family childcare teachers. They provide comprehensive early education services annually to approximately 859 children ages 0–5 years. Their locations in Riverside and Imperial Counties include El Centro, Brawley, Mecca, Calexico and Thermal, which are some of the most underserved counties in California [35].

### 2.3. Community Action Partnership of San Luis Obispo County (CAPSLO) Head Start Program

The CAPSLO early education and childcare program provides high-quality, no-cost childcare, early childhood education and preschool services to low-income families in San Joaquin, Monterey, Kern, Fresno, San Benito, San Luis Obispo, Santa Barbara, Ventura, Orange and Northern San Diego Counties who are eligible for HS/EHS and migrant seasonal HS services based on poverty guidelines published by the federal government [36]. The program serves families with children from birth through age five, with some additional services for pregnant women. The CAPSLO HS program is more than just a daycare or preschool program. It focuses on providing services to the entire family and addressing the developmental needs and health of the whole child. CAPSLO HS helps prepare children for success in school and in life. In addition to quality early learning opportunities and high-quality childcare, children enrolled in HS also receive services that support their physical and emotional health, and parents are provided with the support needed to be a lifelong advocate for their children’s health and wellbeing. CAPSLO assists families with finding their children a medical and dental home or primary care provider and with common barriers to accessing care, including transportation, language translations and costs [36].

### 2.4. Development of the UCLA Oral Health Curriculum

The UCLA pediatric dental residents (with UCLA dental faculty oversight and input from bilingual COHWs who are on staff at UCLA) created a tailored bilingual (English and Spanish) oral health curriculum for rural and migrant/seasonal HS/EHS staff, teachers and parents in collaboration with RCOE and CAPSLO. Multiple meetings were held with the directors of RCOE and CAPSLO to determine what oral health-related topics were of most interest to their organizations based on the needs of their target population. After the initial phone call, UCLA sent the directors of RCOE and CAPSLO a list of several oral health-related topics, including prenatal oral health, caries risk transmission from mother to child, early childhood caries prevention strategies, proper breastfeeding guidelines, milk intake and guidelines regarding sugar-sweetened beverages and healthy snacking, as well as evidence-based information on oral health and hygiene, specifically the universal recommendation of using fluoridated toothpaste for children of all ages. Follow-up phone meetings were dedicated to discussing the importance of each topic and then coming to a consensus on the topics that had the highest priority for RCOE and CAPSLO. The topics chosen for the final version of the oral health education curriculum are presented in Table 1. Once the topics were chosen, the UCLA principal investigator and the directors of RCOE and CAPSLO scheduled the educational workshops. The directors of RCOE and CAPSLO recruited the staff, teachers and parents to participate in the workshops. 

### 2.5. Data Collection

To examine the extent to which pediatric dentists delivered the oral health education material as outlined in the curriculum, process evaluation data were collected using a form that included all nine main topic areas to be covered during the workshop, along with the content for each of the nine main topics. Each topic and content area on the form was assessed using a three-point scale (fully achieved = 1, partially achieved = 2, not achieved = 3). The UCLA project evaluator completed the process evaluation forms for all nine workshops. 

HS/EHS staff, teacher and parent satisfaction with the workshops was assessed using an open forum setting at the end of the workshop, where participants could ask questions and provide feedback. This led to a more informal conversation between the participants and the workshop facilitators. Research has shown that using informal conversations as a qualitative methodology often creates a greater ease of communication that may produce more naturalistic data than other forms of qualitative methods where participants are being audio recorded [37]. The project evaluator documented all the questions and feedback by the participants and summarized the findings. 

A brief open-ended questionnaire was used to assess the UCLA pediatric dental residents’ experience and satisfaction with educating RCOE and CAPSLO teachers, staff and parents. These two open-ended questions were as follows: (1) What were some challenges when leading the workshop? and (2) What was your biggest takeaway when lecturing to an underserved community? The third question asked residents to write a paragraph reflecting on their experience. 

### 2.6. Data Analysis

Data for the process evaluation forms were exported from REDCap into Stata Version 14 (College Station, TX, USA) to conduct univariate analyses.

## 3. Results

### 3.1. Process Evaluation

The process evaluation findings are presented in Table 1. Eight of the nine residents (89%) covered all the required content on topic areas one, two and three (Introduction and Social Determinants of Health, Oral Health, Systemic Health and Diet and Nutrition). Seven of nine residents (78%) covered all the required content on topic area four (Caries Prevention and Management). Five of seven residents (71%) covered all the required content on topic area five (Benefits of Fluoride). Two of four residents (50%) covered all the required content on topic area six (Oral Injuries and Dental Trauma—What to Do). Two of four residents (50%) covered all the required content on topic area seven (Effects of Poor Oral Health on School Performance). One of two residents (50%) covered all the required content on topic area eight (Resuming Brushing in the Classroom). Only two residents discussed topic area nine (Resources for Educators and Families), and they only partially covered the required content. Some residents reported not being able to complete all the required topics because of a lack of time.

### 3.2. Questions on Curriculum Content by Teachers/Staff and Satisfaction with the Workshop

Teachers/staff had several questions for the pediatric dental residents. Please see Table 2 for a list of the questions. The teachers and staff provided positive feedback on the workshop (see Table 2). Overall, they thought the workshop was informative and felt they learned something new about oral health.

### 3.3. Questions on Curriculum Content by Parents and Satisfaction with the Workshop 

Parents participating in the oral health online education workshop had many questions for the pediatric dental residents. Please see Table 3 for a list of the questions. Parents thought the information they received was interesting and useful (see Table 3). 

### 3.4. Pediatric Dental Residents’ Experience with Conducting the Workshops

The pediatric dental residents who conducted the workshops thought it was a great experience and very rewarding (see Table 4 for reflection quotes). Some of the key take-home messages the pediatric dental residents reported regarding lecturing to an underserved community included the following: (1) answer questions in a non-judgmental manner; (2) importance of describing dental trauma to parents; (3) providing parents with information regarding what happens at a dental visit and how they should interpret and follow-up with the recommendations given to them by the dentist; (4) not using too much dental jargon in the presentation to ensure parents are understanding the information being conveyed; (5) importance of covering the basics of oral hygiene (e.g., importance of covering the entire toothbrush with toothpaste), as you cannot always assume parents know all the details of good oral hygiene; and (6) have the presentation be more interactive by starting with a question and answer session to get a sense of what topics the trainees are interested in so the trainer can focus more attention on those topics.

The pediatric dental residents reported the following challenges with leading the workshops: (1) not enough time to discuss all the topics or answer questions; (2) time management (one pediatric dental resident reported it would have been helpful to have the participants complete a short survey to get a sense of what they knew/did not know so more time could be spent on the topics they had the least knowledge on); (3) hard to know how engaged parents are when conducting the training via Zoom; and (4) hard to have an interactive presentation when using Zoom.

## 4. Discussion

The findings from this formative qualitative research study suggest that partnering pediatric dental residents with bilingual (English and Spanish) COHWs to educate HS/EHS teachers, staff and parents on oral health care in rural/migrant areas could result in a rewarding experience for the pediatric dentists that might lead them to practice in these communities upon graduation from their residency program. Additionally, the positive feedback we received from the teachers, staff and parents who participated in the oral health online education workshops indicates they were receptive to receiving oral health education from dental providers who were assisted by bilingual COHWs and found the information they received interesting and useful. 

The process evaluation findings reveal that the pediatric dental residents did not deliver all the topic areas in the curriculum to participants in all the workshops they conducted. The workshops with the teachers and staff were to include all nine topics, while the workshops with the parents were to include topics 1–5, which mainly focused on general oral health information and prevention. However, feedback from the pediatric dental residents showed that they often did not have time to complete all the topics, so they covered topics they felt were most important in regard to key information about oral health education and caries prevention and management, in addition to the topics our RCOE and CAPSLO collaborators thought were most relevant to their target population. This feedback provides valuable information for future workshops to ensure that the oral health education topics and content being delivered is appropriate for each specific target population and can be completed within the time allotted by the organizations we are partnering with. 

The HS/EHS teachers, staff and parents thought the workshops were informative and felt they learned something new about oral health. They had several questions for the pediatric dentists on issues pertaining to dental hygiene, the pros and cons of fluoride, dental trauma, dental treatment options for children, diet and nutrition and pacifier use and thumb sucking. The feedback from the teachers and staff will be used to update and enhance the oral health curriculum for future workshops and, eventually, COHW training with our partners in rural/migrant areas. We were encouraged to see that the workshop participants were receptive to learning about oral health from dental residents who were assisted by COHWs, as previous studies have indicated that migrant populations may be better reached by oral health education programs led solely by CHWs or promotoras [5,6]. For example, a couple of studies have reported that an interactive CHW-led oral health education program was an effective way of increasing oral health knowledge among migrant populations [5,6]. While studies have evaluated dental professional-led oral health education programs in non-migrant populations [38], there is currently a lack of studies examining dental professional-led oral health education programs in migrant populations. Our findings provide an important contribution to the literature on this topic, but more research is needed to compare the effectiveness of oral health education programs led by dental professionals, COHWs or a hybrid model that includes both dental professionals and COHWs.

The pediatric dental residents who conducted the workshops reported their experience as being very rewarding. Based on the reflection paragraphs from the residents, they expressed interest in wanting to have more opportunities to conduct oral health education workshops, as they enjoyed the role of an educator. They provided some key take-home messages regarding lecturing to an underserved community, such as answering questions in a non-judgmental manner and not using too much dental jargon in the presentation to ensure the participants were understanding the information being conveyed. This feedback acknowledges the importance of oral health literacy in terms of making sure the information is being delivered in a culturally and linguistically appropriate manner and at a suitable literacy level for the target population [39,40]. This is where the bilingual COHWs provided invaluable support to the pediatric dentists by helping to ensure the participants understood the information being relayed to them. The pediatric dental residents reported that the main challenge with leading the workshops was not enough time to discuss all the topics or answer questions, as well as the difficulty of giving an interactive presentation when using Zoom. Future workshops/training will be conducted in person. The positive findings by the pediatric dental residents who conducted the workshops lends credence to one of the main objectives of the UCLA School of Dentistry’s Section of Pediatric Dentistry residency program, which is to introduce pediatric dental residents to the many rewards of working in rural/migrant areas. Additional incentives to consider when recruiting dentists to practice in underserved areas might include offering enhanced loan repayment programs and tax benefits [9]. 

There were limitations to this study. Formative research is useful for developing effective strategies for intervention programs, study protocols and survey instruments that are tailored to a specific target population. However, the limitations of formative research include time constraints (which we encountered due to the brief period of time we were allotted for conducting the workshops with parents) and accuracy and generalizability of the information collected, as it is subjective in nature [41,42]. Additionally, the workshops were conducted online using Zoom. While online learning may be more convenient for participants, this format loses the face-to-face aspect of learning, which may result in a lack of connection and engagement between the educator and participants [43]. Future studies should weigh the pros and cons of online vs. in-person oral health education workshops. 

## 5. Conclusions

The findings from this formative research study indicate pediatric dentists who conducted the online oral health workshops in partnership with COHWs thought it was a rewarding experience. The HS/EHS teachers, staff and parents provided positive feedback on the workshops and thought the information they received was interesting and useful. The dual strategy of introducing pediatric dentists to the benefits of working in rural/migrant areas during their residency training and educating individuals from community-based organizations in rural/migrant areas is an innovative and powerful way of building a strong synergistic relationship among dental providers, COHWs and the community, thus leading to increased access to oral care and optimizing oral health in this hard-to-reach population, as well as empowering the community. Additionally, emphasizing the importance of oral health literacy through the use of COHWs and engaging and meeting with HS/EHS teachers, staff and families “where they were” (i.e., many participants joined the oral health education workshops from home or at an HS/EHS facility) proved to be very successful. It is hoped that other pediatric dental residency programs in both the US and non-US countries will consider developing programs to introduce their dental residents/students to the many rewards of educating and working with children and families in rural/migrant areas to help improve the oral health of children living in these underserved areas worldwide. 

## Figures and Tables

**Table 1 ijerph-21-00544-t001:** Process evaluation findings for the 9 workshops *.

Topic Areas Covered in Oral Health Curriculum	Fully Achieved N (%)	Partially Achieved N (%)	Not Achieved N (%)
**1. Introduction and Social Determinants of Health (N = 9)** (ECC disparities, importance of social determinants of health, oral health literacy)	8 (89%)	1 (11%)	0 (0%)
**2. Oral Health and Systemic Health (N = 9)** (What is oral health, mouth anatomy, early childhood caries, caries risk factors, white spot lesions, risks of untreated tooth decay)	8 (89%)	1 (11%)	0 (0%)
**3. Diet and Nutrition (N = 9)** (MyPlate, reading nutrition labels, avoiding sugary beverages, meals and snacks, healthy food/healthy snacks)	8 (89%)	1 (11%)	0 (0%)
**4. Caries Prevention and Management (Age 0–12) (N = 9)** (Oral health in children ages 0–2/oral health in children ages 3–5, how to brush/toothbrushing tips, toothbrushing positions, fluoridated toothpaste, toothbrushes/types of toothbrushes, flossing, thumb sucking and pacifier use)	7 (78%)	2 (22%)	0 (0%)
**5. Benefits of Fluoride (N = 7)** (Importance of fluoride use, why fluoride is helpful, systemic vs. topical fluoride)	5 (71%)	2 (29%)	0 (0%)
**6. Oral Injuries and Dental Trauma—What to do? (N = 4)** (Oral injuries, dental emergencies and what to do)	2 (50%)	2 (50%)	0 (0%)
**7. Effects of Poor Oral Health on School Performance (N = 4)** (Missed school days and delays in performance, signs of oral health problems in children, developmental delays)	2 (50%)	2 (50%)	0 (0%)
**8. Resuming Brushing in the Classroom (N = 2)** (Resuming brushing in the classroom)	1 (50%)	1 (50%)	0 (0%)
**9. Resources for Educators and Families (N = 2)** (Resource kit for families, resource kit for teachers, classroom educator kit)	0 (0%)	2 (100%)	0 (0%)

* Please note that the workshops with the teachers and staff were to include all nine topics when time permitted, while workshops with the parents were only to include topics 1–5, which mainly focused on general oral health information and prevention.

**Table 2 ijerph-21-00544-t002:** Questions about curriculum content from teachers/staff and satisfaction with the workshop.

**Questions on Curriculum Content**
Is plastic or wired floss a better option?
Is a manual or electronic toothbrush better for toothbrushing?
What are the correct techniques for brushing and flossing?
Is fluoride sodium harmful?
Are gummy vitamins harmful for the teeth?
How can staff encourage children to brush their teeth during daycare?
Is it better to floss first or brush your teeth first?
How does chewing gum affect the pH level of your mouth?
If a tooth falls out due to a dental injury, could you reuse it?
If permanent teeth are slightly loose, will they eventually heal?
Participants had many questions and concerns about resuming toothbrushing, especially regarding COVID-19 protocols.
Participants had many questions about how to discuss the use of tap water containing fluoride with parents, as they have been skeptical of tap water in the past.
One participant mentioned that she is interested in dentistry and wanted to know more about scholarship opportunities at UCLA.
**Satisfaction with workshop**
This was a very informative presentation.
Thank you for answering my questions and helping me to better understand oral health.
The presentation was very helpful.
Very nice presentation and thank you to the presenter.
Learned something new today and found it to be very informative.
Presentation was clear and helped me understand the things that affect the teeth.

**Table 3 ijerph-21-00544-t003:** Questions on curriculum content from parents and satisfaction with the workshop.

**Questions on Curriculum Content**
How long do you have to wait to give your child water after brushing their teeth?
Is it necessary to rinse the teeth after brushing?
At what age can a child use mouthwash?
How often is it recommended to put fluoride varnish on children’s teeth?
How do you prepare a child for their dental visit?
What are the different dental treatment options for children (e.g., sedation, IVGA, etc.)?
What is an appropriate liquid to put into a baby bottle besides water?
At what age should you start flossing your child’s teeth?
How often should toothbrushes be changed in general?
Should toothbrushes be replaced when a child gets sick?
Are strong teeth and gums hereditary?
At what age do children lose their baby teeth?
Parents had questions about pacifier use and thumb sucking (e.g., they wanted to know what causes natural spacing between baby teeth).
**Satisfaction with workshop**
The presentation was very clear, and the doctor did a great job on answering the questions.
The dentist delivered a great presentation with useful information.
Learned something new and plan to apply it at home.
The session was very interesting and helpful.
The lecture was clear and helped us understand the different ways to protect my child’s teeth.
The presentation was very interesting.
The presenter did a good job and was very clear with delivering the information.

**Table 4 ijerph-21-00544-t004:** Reflection quotes from pediatric dental residents who conducted the workshops.

“It was a great experience as this was the first time I presented for so many people and had a whole hour for the presentation. It was good to cover so many important topics and people had great questions at the end and nice comments.”
“I enjoyed giving this presentation and preparing for it. I had given presentations in Spanish in the past, but it had been several years, and I had forgotten how meaningful it can be. I hope there is opportunities in the future to do them in person because it would be easier to connect with them and assess how much or what they know about the topics we are presenting. My favorite part was at the end when parents got to ask questions. It is interesting to hear and address their concerns, but it also gives us information about their struggles and how to come up with different strategies to help them through them.”
“I enjoyed giving this presentation, this time it seemed like there were more parents and more questions at the end. Reflecting back, if I have another opportunity to present again, I would make the presentation even more interactive so there is more opportunity for them to ask questions and discuss more topics than the ones being presented. Prevention topics are definitely very important but based on the questions they had, they need more information about what happens after kids have dental needs, what dental texts are recommended and why. I think this would better help them make informed decisions for their children’s dental health.”
“The lecture went well; parents shared some personal stories about their children’s dental trauma and dental experience. It was a productive lecture to go over a variety of topics to educate teachers, parents, and staff that play a significant role in the child’s oral health.”
“My participation with the SLO parents was very easy during the Zoom session. I appreciated the COHWs help. I could not see the community or people I was presenting to which made speaking to them a bit less intimate and I didn’t feel the connection as much. However, the process was still quite rewarding because I hope I was able to clarify a few things and present helpful knowledge to the parents that took the time to attend in person to the event. I love being in an educating role, so this was right up my alley.”
“The lecture was a great means to teach and connect with parents. I answered questions thoroughly and dispelled misconceptions regarding fluoride. I conveyed valuable information regarding having a dental home and its importance when it comes to prevention of caries and trauma injury.”

## Data Availability

To obtain access to the datasets and survey instruments used in this study, please contact Janni Kinsler at jnhaiem@dentistry.ucla.edu.

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
