# Peer review of "Pediatric Oral Health Online Education for Rural and Migrant Head Start Programs in the United States"

_ijerph, 2024, doi:10.3390/ijerph21050544_

Round 1

Reviewer 1 Report

Comments and Suggestions for Authors The proposed idea is very interesting,
but presented in an unscientific way.
Each question should correspond to a
comparable percentage or numerical value, such that it can be objective.
In the work the only numerical data relate to an overall judgment of
the 9 workshops, the real level of knowledge acquired by the categories
of people analyzed cannot be quantified.
In this work there are methodological problems of data collection.
The knowledge acquired should be evaluated based on the number of
correct answers given after the training program compared to the total,
with a robust statistical investigation.
Introduction and bibliography are fine, but the Materials and Methods
and Results section should be much more rigorous.

Author Response

Comments and Suggestions for Authors (Reviewer 1)

Comment 1: The proposed idea is very interesting but presented in an unscientific way.  Each question should correspond to a comparable percentage or numerical value, such that it can be objective.

Response: Given the focus of the study was to assess whether staff, teachers and families from rural/migrant Head Start programs in California were receptive to oral health online education workshops conducted by pediatric dental residents who were assisted by bilingual (English Spanish) COHWs and not to assess participant knowledge, we chose to conduct a Q&A session where participants could ask any questions they had. This led to a more informal conversation between the participants and the workshop facilitators. Using informal conversations as a qualitative methodology has been shown to create a greater ease of communication that oftentimes produces more naturalistic data than other forms of qualitative methods where participants are being recorded (Swain J, King B. Using informal conversations in qualitative research. International Journal of Qualitative Methods. 2022;21:1-10. DOI: 10.1177/16094069221085056). The formative qualitative data obtained in this study will be used to enhance the oral health education workshops for a recent grant that was funded which will include conducting formal focus groups and pre-posttest knowledge questionnaires with workshop participants in rural/migrant populations.

Comment 2: In the work, the only numerical data relate to an overall judgment of the 9 workshops, the real level of knowledge acquired by the categories of people analyzed cannot be quantified. In this work there are methodological problems of data collection. The knowledge acquired should be evaluated based on the number of correct answers given after the training program compared to the total, with a robust statistical investigation. Materials and Methods
and Results section should be much more rigorous.

Response:  Please see response to Comment 1 above.

Reviewer 2 Report

Comments and Suggestions for Authors

Please take the following aspects into consideration:

- In the methodology, they could explain in more detail the methods used for the workshop and not just the topics covered.

- Table 1: Could be better presented. There are sections without any mention of "n" or percentages even though they are "zero". The title says 9 workshops and the "n" do not add up to 9. I would say I was confused trying to understand.

- Table 2: Why talk about toothpaste brands... it is not necessary to include them in the table (Colgate or other). Please remove the brand of toothpaste mentioned in the workshop so as not to mislead readers, it is not necessary to mention it in this article.

- It would be interesting to include photos from the workshops.

Author Response

Comments and Suggestions for Authors (Reviewer 2)

Comment 1: In the methodology, they could explain in more detail the methods used for the workshop and not just the topics covered.

Response: We have provided more detail on the methods used for the workshops (please see page 4, lines 192-194 and page 5, lines 205-209).

Comment 2: Table 1 could be better presented. There are sections without any mention of "n" or percentages even though they are "zero". The title says 9 workshops and the "n" do not add up to 9. I would say I was confused trying to understand.

Response: Table 1 has been revised (please see page 5, line 217). We have included “zero percent” where appropriate. We also provide a footnote explaining why all n’s do not add up to 9. Additionally, we provided a more detailed discussion of the process evaluation findings in the Discussion section explaining why all n’s do not add up to 9 (please see page 9, lines 301-305).

Comment 3: Table 2: Why talk about toothpaste brands... it is not necessary to include them in the table (Colgate or other). Please remove the brand of toothpaste mentioned in the workshop so as not to mislead readers, it is not necessary to mention it in this article.

Response: We deleted the question that referred to the toothpaste brand, Colgate (please see page 6, lines 249-250).

Comment 4: It would be interesting to include photos from the workshops.

Response: We agree it would be interesting to include photos from the workshops, but this would not have been possible due to IRB constraints.

Reviewer 3 Report

Comments and Suggestions for Authors

Dear Authors, 

my comments are listed below. 

Title: add "online" in front of education (ad everywhere else in the text)

Results have to be more clear: in Tables 2. , 3. and 4. results have to be more specific - add percentage of participant who asked certain questions. 

You do not have to repeat questions in the text as they are presented in Tables. 

Discussion: 

-add similar studies and their results (not necessary from USA).  

- add in limitations that this was an online education and potential benefit of non online education in the future.  

Author Response

Comments and Suggestions for Authors (Reviewer 3)

Comment 1: Title: add "online" in front of education (add everywhere else in the text).

Response: We added “online” in front of education in the title and throughout the text.

Comment 2: Results have to be clearer: in Tables 2, 3. and 4. results have to be more specific - add percentage of participant who asked certain questions. 

Response: Given the focus of the study was to assess whether staff, teachers and families from rural/migrant Head Start programs in California were receptive to oral health online education workshops conducted by pediatric dental residents who were assisted by bilingual (English Spanish) COHWs and not to assess participant knowledge, we chose to conduct a Q&A session where participants could ask any questions they had without them feeling like they were being “evaluated.” This led to a more informal conversation between the participants and the workshop facilitators. Using informal conversations as a qualitative methodology has been shown to create a greater ease of communication that oftentimes produces more naturalistic data than other forms of qualitative methods where participants are being recorded (Swain J, King B. Using informal conversations in qualitative research. International Journal of Qualitative Methods. 2022;21:1-10. DOI: 10.1177/16094069221085056). The formative qualitative data obtained in this study will be used to enhance the oral health education workshops for a recent grant that was funded which will include conducting formal focus groups and pre-posttest knowledge questionnaires with workshop participants in rural/migrant populations.

Comment 3: You do not have to repeat questions in the text as they are presented in Tables. 

Response: We deleted questions from the text that were also included in the Tables (please see page 6, lines 241-246 and page 7, lines 254-260).

Comment 4: Discussion: Add similar studies and their results (not necessary from USA). Add in limitations that this was an online education and potential benefit of non-online education in the future. 

Response: We have revised the discussion section to include findings from similar studies. We also discuss the potential pros and cons of online education in the limitations section (please see page 9, lines 319-327 and page 10, lines 354-359). 

Round 2

Reviewer 3 Report

Comments and Suggestions for Authors

*